# RECTIFIED ATTRIBUTE-MISSING GRAPH CLUSTERING

## ABSTRACT

Deep Graph Clustering(DGC) has gained widespread attention in some tasks like face recognition and social network analysis because of its powerful capability of capturing the latent distribution of multi-view graph-structured data and grouping nodes in the graph into different clusters. However, in the real world, they often face the situation where attributes of some nodes are missing, resulting in clustering performance degradation. Although many methods have been developed to mitigate the problem, most of them are two-stage methods that separate embedding learning from clustering, and may cause deviation of node embedding because of the missing nodes. To address this problem, we propose a novel end-to-end attribute-missing graph clustering learning method termed $\underline{R}$ectified $\underline{A}$ttribute-$\underline{M}$issing $\underline{G}$raph $\underline{C}$lustering (R-AMGC). First, it performs two augmentations to generate two attribute views, and the missing attributes are set to be learnable in one of the views. Subsequently, we maximize the mutual information between two encoded views via contrastive learning and then effectively mitigate the embedding distortion. Additionally, to learn clustering-friendly embedding, we design a module termed triple constraint, which not only maintains the alignment between graph structure and cluster assignment but also captures structural and attribute information. Extensive experiments on four graph datasets have strongly validated the effectiveness and superiority of R-AMGC compared to other counterparts.

## 1 INTRODUCTION

Deep Graph Clustering (DGC) is an unsupervised method designed to put similar nodes into the same group while effectively pushing farther those belonging to different categories within graph-structured data. Taking advantage of the strong representation capability of deep learning (LeCun et al., 2015), DGC has witnessed fruitful advances and has also been put into practical situations, such as community detection (Park et al., 2022). However, in real-world applications, DGC often faces the challenge of missing data due to issues such as privacy regulations or data corruption. This gives rise to the problem of Attribute-Missing Graph Clustering, namely AMGC, which has increasingly become critical in the field of DGC.

In the face of the attribute missing scenario, some earlier works, such as Node2Vec Grover & Leskovec (2016), High-Order Proximity preserved Embedding(HOPE) Ou et al. (2016), and Learning Graph Representations with Global Structural Information(GraRep) Cao et al. (2015), only utilizes network topology while ignoring rich attribute information. Many researches like Graph attention network(GAT) Velickovic et al. (2017), GCN Kipf (2016), GraphSAGE Hamilton et al. (2017) that are supposed to apply to DGC with complete attributes, are also used to handle the situation where the attributes are incomplete on graph nodes. Because these networks have the natural trait that they can aggregate neighbor node information when they are working, we can gain a relatively reliable embedding via these networks. However, these methods have limitations in effectively recovering missing attributes in a situation of an extremely high missing rate, resulting in poor performance.

Recently, many advancements have been made to exclusively deal with the problem of attribute-missing graph learning, but there are often two critical flaws in these methods. One is that some of

them are based on Feature Propagation(FP) Rossi et al. (2022), like Confidence-based feature imputation for graphs with partially known features(PACI) Um et al. (2023), AttriReBoost(ARB) Li et al. (2025b), and are imputation-first, which means the training process for these methods is to carry out imputation first on the original missing data and then put the imputed data into the network. These methods may distort the node embedding in the latent space (Li et al., 2025a). The other is that these methods are not only for AMGC. Specifically, these methods, such as Multi-view collaborative for graph attribute imputation(MOBA) Yu et al. (2024), Variational Graph Auto-Encoders(VAGE) Kipf & Welling (2016), and Incomplete Graph Learning via Attribute-Structure Decoupled Variational Auto-Encoder(ASD-VAE) Jiang et al. (2024), can handle the attribute-missing situation, but they are not designed for AMGC, which means that it is impossible to achieve end-to-end training for clustering. The accumulation of sub-task errors can limit the final performance (Tu et al., 2024).

To address the limitations above, we propose a novel method designed exclusively for AMGC termed Rectified Attribute-Missing Graph Clustering(R-AMGC). The process of the proposed methods consists of three parts. In the first part, we apply graph augmentation to generate two attribute views(the unknown attributes are set as learnable parameters for the first view).To mitigate the problem of embedding distortion caused by missing nodes, the main idea of the second part is to decrease the adverse effect of missing nodes by applying the multi-head attention mechanism. In the third part, we decode the rectified embedding to restore structure and attribute information and use the clustering consistency loss to ensure the consistency between graph structure and clustering results, thus forming an end-to-end learning. In summary, the contributions of the proposed method can be summarized as follows:

- We design the module embedding rectification based on a multi-head attention mechanism. The key function of the module is to rectify the distorted node embedding by quantifying the contribution of missing nodes and applying a decay coefficient to diminish the effects of missing nodes.

- We introduce R-AMGC, a novel unified optimization learning framework for AMGC, where we design the triple constraint to learn clustering-friendly embedding and achieve high-quality attribute imputation by recovering attribute and topological information.

- Extensive experiments on four graph datasets have solidly demonstrated the effectiveness and superiority of R-AMGC compared to other competitors.

## 2 RELATED WORK

### 2.1 DEEP GRAPH CLUSTERING

Some works develop reconstructive methods that force the network to encode the features in the graph and then recover the graph information from the learned embeddings. Thus, reconstructive methods such as the Structural Deep Clustering Network(SDCN) Bo et al. (2020), Deep Fusion Clustering Network(DFCN) Tu et al. (2021), Rethinking graph auto-encoder models for attributed graph clustering(R-GAE) Mrabah et al. (2022), and Deep attention-guided graph clustering with dual self-supervision(DAGC) Peng et al. (2022) focus on the intra-data information in the graph and adopt the node attributes and graph structure as self-supervision signals.

Some methods based on adversarial learning, such as Adversarial Graph Embedding for ensemble clustering(AGAE) Tao et al. (2019), Learning graph embedding with adversarial training methods(ARGA) Tao et al. (2019), Wasserstein adversarially regularized graph autoencoder(WARGA) Liang & Gao (2023), aim to improve the quality of features by creating a zero-sum game between the generator and the discriminator. Specifically, the discriminator is trained to recognize whether learned features are from the real data distribution, while the generators are designed to generate confusing embeddings to cheat the discriminator.

Contrastive learning is also often applied to the field of DGC to improve the discriminativeness of features by pulling together the positive samples while pushing away the negative ones. Thus, contrastive methods such as Contrastive multi-view representation learning on graphs(MVGRL) Hassani & Khasahmadi (2020), Graph debiased contrastive learning with joint representation clustering(GDCL) Zhao et al. (2021), Towards unsupervised deep graph structure learning(SUBLIME)

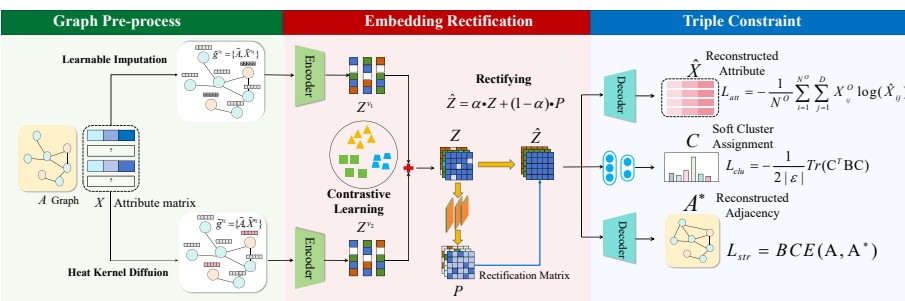

Figure 1: Overview of R-AMGC. The overall architecture of R-AMGC. R-AMGC first generates two attribute views with local and global structure information, respectively. Then, the information sources under these two views are encoded into node embeddings by Siamese encoders. The Embedding Rectification makes the node embedding more distinguishable from one another and carries out a rectifying operation on them to obtain a semantically rich embedding. Finally, we apply the Triple Constraint to lead the network to high-quality attribute imputation and maintain alignment between graph structure and cluster assignment.

Liu et al. (2022) focus on the inter-data information and construct the self-supervision signals via meaningful relationships between samples.

## 2.2 ATTRIBUTE-MISSING GRAPH LEARNING

These methods leverage the information on observed nodes and edges, combining advanced graph learning techniques and data imputation methods to impute missing attributes.

The goal of some of these methods is to generate features or latent representations for nodes that lack attributes. Structure-Attribute Transformer(SAT) Chen et al. (2020), a GNN-based Structure-Attribute Transformer method, imputes missing data by distribution matching in the latent space. A new attribute-missing network embedding approach(Amer) Jin et al. (2022), a game-theoretic GNN method, carries out imputation by combining game theory principles with mutual information optimization in the unified learning framework.However, generative adversarial strategies may have a problem capturing complex connections between nodes.

Based on the above analysis, a novel imputation strategy, Initializing Then Refining (ITR) Tu et al. (2022), was proposed, which combines reliable node attributes with structural information from the graph to generate a relatively correct embedding for attribute-missing nodes. Attribute-missing Graph Contrastive Learning (AmGCL) Zhang et al. (2023) integrates node attribute completion and graph representation learning into a unified contrastive learning model. In particular, AmGCL introduces a novel combination of Dirichlet energy minimization and contrastive learning methods. Structured Variational Graph Autoencoder (SVGA) Yoo et al. (2022), which uses structured variational inference to impose constraints and regularization on the latent variable distribution. However, these methods fail to integrate the structure and attribute information effectively during the encoding process.

## 3 METHODOLOGY

In this section, we introduce the Rectified Attribute-Missing Graph Clustering (R-AMGC), a novel approach aimed at enhancing graph clustering with incomplete data. The R-AMGC framework is shown in Fig. 1.

## 3.1 NOTATIONS

**Notations** Let $\mathcal{G} = \{\mathbf{X}, \mathbf{A}, \mathcal{E}\}$ represent an undirected graph, consider $\mathbf{V} = \{v_1, v_2, ..., v_N\}$ be a set of $N$ nodes classified into $K$ classes, and $\mathcal{E}$ be a set of edges.$E$ The number of edges $\mathbf{X} \in \mathbb{R}^{N \times D}$ denotes the attribute matrix of all nodes, where $D$ is the dimension of attributes. $\mathbf{A} \in \{0, 1\}^{N \times N}$ denote the adjacency matrix, where $a_{ij} = 1$ if $(v_i, v_j) \in \mathcal{E}$. With the renormalization trick $\bar{\mathbf{A}} = \mathbf{A} + \mathbf{I}_N$ in GCN (Kipf, 2016), where $\mathbf{I}_N$ is the N-order identity matrix, the normalized adjacency matrix is denoted as $\tilde{\mathbf{A}} = \hat{\mathbf{D}}^{-1/2} \bar{\mathbf{A}} \hat{\mathbf{D}}^{-1/2}.\hat{\mathbf{D}} = diag(\hat{d}_1, \hat{d}_2, ..., \hat{d}_N) \in \mathbb{R}^{N \times N}$ is the degree matrix, and $\hat{d}_i = \sum_{j=1}^{N} \bar{\mathbf{A}}_{i,j}$. $\gamma_{ij} = \frac{1}{\sqrt{\hat{d}_i \cdot \hat{d}_j}}$ is the normalization coefficient assigned to the edge$(v_i, v_j)$ .

The modularity matrix $\mathbf{B}$ can be defined as $\mathbf{B} = \mathbf{A} - \frac{\hat{\mathbf{d}} \hat{\mathbf{d}}^{\mathbf{T}}}{2|\epsilon|}$ a measure of how similar the nodes are in the graph.$\hat{\mathbf{d}}$ is the degree vector. The symmetric normalized graph Laplacian matrix is defined as $\tilde{\mathbf{L}} = \mathbf{I}_N - \tilde{\mathbf{A}}$. The missing feature mask $\mathbf{M} \in \{0, 1\}^{N \times D}$ is defined such that $\mathbf{M}_{i,j} = 1$ indicates that the $j$-th feature of node $i$ is missing, while $\mathbf{M}_{i,j} = 0$ signifies that the feature is observed.

## 3.2 GRAPH PRE-PROCESS

Before training the network, we first generate two augmented views derived from the original attribute matrix $X$ that contain unknown node attributes. For the first view, inspired by **MATE** Peng et al. (2024), we set the missing value as a learnable parameter that is randomly generated and follows a Gaussian distribution to obtain the attribute matrix $\mathbf{X}^1$. Next, to improve the local context information, we perform the operation as follows:

$$\tilde{\mathbf{X}}^1 = \tilde{\mathbf{A}} \mathbf{X}^1 \tag{1}$$

For the second view, we need to initialize the missing value as zero to get $\mathbf{X}^2$ , then perform a common technique for augmentation in graphs, termed Heat Kernel Diffusion (Hassani & Khasahmadi, 2020). The operation is as follows:

$$\tilde{\mathbf{X}}^2 = \hat{\mathbf{A}} \mathbf{X}^2 \tag{2}$$

where $\hat{\mathbf{A}} = \exp(\beta \tilde{\mathbf{L}})$ denotes the diffusion matrix. The $\beta$ denotes the temperature parameters that can control the diffusion intensity, and is set as 0.1 in this case.

The diffusion matrix $\hat{\mathbf{A}}$ contains global structure information, so the attributes in $\tilde{\mathbf{X}}^{\mathbf{2}}$ are actually the aggregation of the neighboring nodes' information and higher-order neighboring nodes. However, the attribute features of the known nodes are actually distorted. Therefore, we need to replace these distorted but known node attributes with observed ones. Specifically, we perform the operation as follows :

$$\tilde{\mathbf{X}}^{v_1} = \tilde{\mathbf{X}}^1 \odot \mathbf{M} + \mathbf{X} \odot (\mathbf{1} - \mathbf{M}) \tag{3}$$

$$\tilde{\mathbf{X}}^{v_2} = \tilde{\mathbf{X}}^2 \odot \mathbf{M} + \mathbf{X} \odot (\mathbf{1} - \mathbf{M}) \tag{4}$$

where $\odot$ and $\mathbf{1} \in \mathbb{R}^{N \times N}$ denote element-wise multiplication and an all-ones matrix, respectively. Finally, we define two views $\tilde{\mathcal{G}}^{v_1} = \{\tilde{\mathbf{X}}^{v_1}, \tilde{\mathbf{A}}\}$ $\tilde{\mathcal{G}}^{v_2} = \{\tilde{\mathbf{X}}^{v_2}, \tilde{\mathbf{A}}\}$ and take them as input for the graph encoding.

## 3.3 EMBEDDING RECTIFICATION

In this work, we apply a graph neural network-based siamese architecture as an encoder. Subsequently, inspired by the multi-head attention mechanism (Vaswani et al., 2017), we design the module embedding rectification to improve the discriminativeness of the node embedding and rectify the deviation of node embedding.

### 3.3.1 NODE EMBEDDING

The encoder adopts a common graph convolutional network. $f_e(\cdot)$ Then, the encoder $f_e(\cdot)$ with two layers GCN extracts the graph embeddings for views $\tilde{\mathcal{G}}^{v_1}$ and $\tilde{\mathcal{G}}^{v_2}$:

$$\mathbf{Z}^{v_1} = f_e(\tilde{\mathcal{G}}^{v_1}) = \sigma(\tilde{\mathbf{A}} \sigma(\tilde{\mathbf{A}} \mathbf{X}^{v_1} \mathbf{\Theta}^1) \mathbf{\Theta}^2) \tag{5}$$

$$\mathbf{Z}^{v_2} = f_e(\tilde{\mathcal{G}}^{v_2}) = \sigma(\tilde{\mathbf{A}} \sigma(\tilde{\mathbf{A}} \mathbf{X}^{v_2} \mathbf{\Theta}^1) \mathbf{\Theta}^2) \tag{6}$$

where $\Theta^1$ $\Theta^2$ are the learnable parameters of the network $f_e(\cdot)$, $\sigma(\cdot)$ represents the sigmoid activation function, and $\mathbf{Z}^{v_1}, \mathbf{Z}^{v_2} \in \mathbb{R}^{N \times \bar{D}}$ the dimension of node embeddings represents the node embedding matrix in two views.

### 3.3.2 CONTRASTIVE LEARNING

To ensure consistency and complementarity of attribute information between two views $\mathbf{Z}^{v_1} \mathbf{Z}^{v_2}$, a contrastive loss function is applied to maximize the mutual information of node embeddings under different views, which can be expressed as:

$$\mathcal{C}(\mathbf{Z}^{v_1}, \mathbf{Z}^{v_2}) = \sum_{i=1}^{N} \log \frac{\exp\left(\text{sim}(\mathbf{z}_i^{v_1}, \mathbf{z}_i^{v_2})/\tau\right)}{\sum_{j=1, j\neq i}^{N} \exp\left(\text{sim}(\mathbf{z}_i^{v_1}, \mathbf{z}_j^{v_2})/\tau\right)} \tag{7}$$

$$\mathcal{L}_{con} = -\frac{1}{2N}(\mathcal{C}(\mathbf{Z}^{v_1}, \mathbf{Z}^{v_2}) + \mathcal{C}(\mathbf{Z}^{v_2}, \mathbf{Z}^{v_1})) \tag{8}$$

### 3.3.3 RECTIFYING STAGE

Although node embedding is more distinguishable from one another through contrastive learning, the problem still exists that the GCN-based encoding process causes the deviation of the node embedding due to the missing nodes. To address this issue, the part aims to reduce the effects of missing nodes throughout the process of training.
Before the denoising stage, we carry out the fusion operation on $\mathbf{Z}^{v_1}$ and $\mathbf{Z}^{v_2}$ to get $\mathbf{Z}$ as follows:

$$\mathbf{Z} = (\mathbf{Z}^{v_1} + \mathbf{Z}^{v_1})/2 \tag{9}$$

To mitigate the impact of uneven node degree distribution in the graph on feature aggregation, it is necessary to calculate the normalization coefficients based on node degrees. The normalization coefficient can be computed as follows:

$$\gamma_{ij} = \frac{1}{\sqrt{\hat{d}_i \cdot \hat{d}_j}} \tag{10}$$

To map node embedding to attention scores through the attention projection layer, we calculate attention weights based on the difference between the attention scores of the start and end nodes of edges, and optimize them by combining normalization coefficients as follows:

$$w_h(i, j) = \frac{\sigma(\mathbf{z}_i \mathbf{W}_h - \mathbf{z}_j \mathbf{W}_h)}{\gamma_{ij}} \tag{11}$$

where $w_h(i, j)$ is the attention weight of the edge $(v_i, v_j)$ in the $h\_th$ head $\mathbf{W}_h \in \mathbb{R}^{D \times 1}$ is the projection weight of the $h$-th attention head.

For each attention head, we weight the features of the end nodes of edges using attention weights and aggregate them according to the indices of the start nodes of edges.

$$\mathbf{F}_h = \mathbf{W}_{h,E} \odot \bar{\mathbf{Z}} \tag{12}$$

where $\mathbf{F}_h \in \mathbb{R}^{E \times D}$ denotes the weighted feature matrix of the $h$-th head. $E$ denotes the number of edges. $\mathbf{W}_{h,E} \in \mathbb{R}^{E \times 1}$ is the edge attention weight matrix of the $h$-th head (a column vector composed of $w_h(i, j)$). $\bar{\mathbf{Z}} \in \mathbb{R}^{E \times D}$ denotes the end node embedding matrix. $\bar{\mathbf{z}}_k = \mathbf{z}_j$ for the k-th edge $(v_i, v_j)$. Note that the equation above is performed based on the broadcasting mechanism.

$$\mathbf{G}_h = \sum_{(i,j)\in\epsilon} \mathbf{F}_h(i, j) \tag{13}$$

Subsequently, we can split the aggregated feature $\mathbf{G}_h$ into contributions from known nodes and missing nodes. A decay coefficient is applied to the contributions of missing nodes to mitigate interference from missing nodes. The whole process can be formulated as follows:

$$\mathbf{G}_h^c = \mathbf{G}_h \odot \mathbf{m} + \lambda \cdot \mathbf{G}_h \odot (\mathbf{1}_n - \mathbf{m}) \tag{14}$$

where $\mathbf{m} \in \mathbb{R}^{N \times 1}$ and $\mathbf{1}_n \in \mathbb{R}^{N \times 1}$ denote the mask vector of all nodes and the all-ones vector, respectively, and $m_i = 1$ when the i-th node is observed. The first term and the second term signify the contribution from known nodes and missing nodes, respectively. $\lambda$ denotes the decay coefficient (set to 0.3 in the paper) for missing-node contributions.

$$\mathbf{G} = \frac{1}{H} \sum_{h=1}^{H} \mathbf{G}_h^c \tag{15}$$

where $\mathbf{G} \in \mathbb{R}^{N \times D}$ represents the feature matrix after multi-head attention aggregation and adjustment. H denotes the number of the head.

Next, we perform a linear transformation on $\mathbf{G}$ through the feature propagation layer(one-layer MLP), and then combine it with a residual connection to obtain the rectification matrix $\mathbf{P}$. We define the projection operation mentioned above as $Proj(\cdot)$, then the rectification matrix is obtained as follows:

$$\mathbf{P} = Proj(\mathbf{G}) \tag{16}$$

After all the steps above, we can finally get the rectified embedding $\hat{\mathbf{z}}$ as follows:

$$\hat{\mathbf{Z}} = \alpha \cdot \mathbf{Z} + (1 - \alpha) \cdot \mathbf{P} \tag{17}$$

Where $\alpha$ denotes the rectification coefficient(set to 0.5 in the paper) To sum up, the module Embedding Rectification mitigates the embedding distortion because of the GCN-based aggregation mechanism by reducing the contribution of missing nodes to their neighboring nodes.

### 3.4 TRIPLE CONSTRAINT

To guide the network to perform high-quality imputation and learn clustering-friendly embedding, we design the module termed triple constraint, including three key loss functions.

#### 3.4.1 STRUCTURE RECONSTRUCTION LOSS

We can decode the denoised node embedding $\hat{\mathbf{Z}}$ to reconstruct the original adjacency matrix by using the inner product decoder, and then apply the binary cross-entropy function between the reconstructed adjacency matrix and the original adjacency matrix as follows:

$$\mathcal{L}_{str} = -\frac{1}{|\epsilon|} \sum_{(i,j) \in \epsilon} [\mathbf{A}_{i,j} \cdot \log(\sigma(\hat{\mathbf{z}}_i \cdot \hat{\mathbf{z}}_j^T)) \ + \ (1 - \mathbf{A}_{i,j}) \cdot \log(1 - \sigma(\hat{\mathbf{z}}_i \cdot \hat{\mathbf{z}}_j^T))] \tag{18}$$

where $\mathbf{A}_{i,j}$ denotes the elements in the original adjacency matrix. We can restore the structural information by minimizing the reconstruction structure loss.

#### 3.4.2 ATTRIBUTE RECONSTRUCTION LOSS

We can decode the node embedding $\hat{Z}$ through a simple decoder $f_{\theta_1}(\cdot)$ (one-layer MLP) to recover the original attribute information. Specifically, the operation is as follows:

$$\mathcal{L}_{att} = -\frac{1}{N^o} \sum_{i=1}^{N^o} \sum_{j=1}^{\bar{D}} \mathbf{X}_{i,j}^o \log(\hat{\mathbf{X}}_{i,j}) \tag{19}$$

where $\hat{\mathbf{X}} = f_{\theta_1}(\hat{\mathbf{Z}})$ represents the reconstructed attribute matrix, and $\hat{\mathbf{X}}^o$ represents the attribute-observable part of the reconstructed attribute matrix.

#### 3.4.3 CLUSTERING CONSISTENCY LOSS

To ensure the consistency between the graph structure and the clustering results, we apply a fully differentiable unsupervised clustering objective loss, which optimizes soft cluster assignments to control for inhomogeneities in the graph(Tsitsulin et al., 2023). The loss function can be formulated as follows:

$$\mathcal{L}_{clu} = -\frac{1}{2|\epsilon|}Tr(\mathbf{C^T BC}) \tag{20}$$

where here $\mathbf{C} \in R^{N \times K}$ and $\mathbf{B} \in \mathbb{R}^{N \times N}$ denote soft cluster assignment and the modularity matrix, respectively, $(\cdot)$ represents the calculation for the trace of a matrix. $softmax(\cdot)$ and $f_{\theta_2(\cdot)}$ denote the normalized activation function and one-layer MLP, respectively. Minimizing the loss function can push nodes that are close in the graph into the same cluster.

The overall objective of $R - AMGC$ consists of four parts:

$$\mathcal{L} = \mathcal{L}_{con} + \mathcal{L}_{str} + \mathcal{L}_{att} + \mathcal{L}_{clu} \tag{21}$$

## 4 EXPERIMENTS

In this section, to validate the effectiveness and the superiority of the proposed R-AMGC, we conduct several experiments.

### 4.1 EXPERIMENTAL SETUP

**Datasets.** We experimented with four benchmark datasets from two different domains: citation networks (Cora, CiteSeer)(Sen et al., 2008), and recommendation networks (Amazon-Computers and Amazon-Photo )(Shchur et al., 2018).

**Comparing Methods** To showcase the effectiveness of the proposed R-AMGC, we conduct comparisons against state-of-the-art attribute-complete deep graph clustering, including **DAEGC** Wang et al. (2019), **GDCL** Zhao et al. (2021), and **DIMVC** Xu et al. (2022), and state-of-the-art attribute-incomplete deep graph learning methods, including **SAT**Chen et al. (2020), **SVGA** Yoo et al. (2022), **ITR** Tu et al. (2022), **MATE** Peng et al. (2024), and **AMGC** Tu et al. (2024).

**Implementation Details** The experiments are conducted on a system equipped with an Intel Core i7-14650HX CPU, an NVIDIA GeForce RTX 4060 GPU, and 16GB of RAM. All experiments are implemented using the PyTorch platform. The maximum number of training epochs is set to 200.

**Evaluation Metrics** To evaluate clustering performance, we use four standard metrics: Accuracy (ACC), Normalized Mutual Information (NMI), Average Rand Index (ARI), and Macro F1-score (F1). Higher values reflect better clustering outcomes.

### 4.2 COMPARISON RESULTS

In Table 1, we comprehensively compare our proposed method with several state-of-the-art methods that have shown superior performance in recent years. We conducted a performance evaluation of R-AMGC on four benchmark datasets, considering three levels of missing data: [0.3, 0.6, 0.9]. To emphasize the results, the best and sub-optimal performance are indicated by bold and underlined formatting, respectively.

### 4.3 ABLATION STUDIES

In this section, to verify the effectiveness of each module and investigate how the two modules, embedding rectification and triple constraint, influence the performance, specifically, we conduct ablation studies on the embeddding rectification, Structure Reconstruction Loss $\mathcal{L}_{str}$, Attribute Reconstruction Loss $\mathcal{L}_{att}$, and Clustering Consistency Loss $\mathcal{L}_{clu}$. Furthermore, we use 'Re' to represent embedding rectification. '$\checkmark$ 'and '$\times$' represent whether the corresponding methods are used or not, respectively. As shown in Table 2, the best performance can be achieved when all modules are considered.

### 4.4 PERFORMANCE WITH DIFFERENT MISSING RATES

To further validate the robustness and effectiveness of R-AMGC, it is crucial to evaluate how well the method performs under varying levels of missing attributes. This was done by comparing R-AMGC with three baseline methods, SVGA, ITR, and MATE, across four benchmark datasets:

Table 1: The node clustering performance for four benchmark datasets is evaluated using four metrics. "OOM" means out-of-memory. The best results are highlighted in bold, while the second-best are underlined.

| Missing rate | | 0.3 | | | | 0.6 | | | | 0.9 | | |
|---|---|---|---|---|---|---|---|---|---|---|---|---|
| Method | ACC | NMI | ARI | F1 | ACC | NMI | ARI | F1 | ACC | NMI | ARI | F1 |
| **CORA** DAEGC | 54.93 | 36.85 | 30.44 | 48.88 | 48.60 | 29.93 | 24.27 | 44.08 | 33.57 | 13.16 | 8.64 | 31.13 |
| GDCL | 66.84 | 48.66 | 40.82 | 58.17 | 52.42 | 35.52 | 32.36 | 46.82 | 32.18 | 11.83 | 3.21 | 18.33 |
| DIMVC | 63.72 | 47.46 | 39.31 | 55.24 | 44.11 | 24.22 | 16.61 | 38.45 | 31.11 | 6.34 | 2.35 | 16.39 |
| SAT | 68.72 | 49.60 | 45.03 | 61.83 | 60.11 | 44.16 | 36.31 | 53.14 | 33.23 | 10.04 | 4.14 | 20.35 |
| SVGA | 64.14 | 51.11 | 41.06 | 60.83 | 44.35 | 29.18 | 17.33 | 35.12 | 27.29 | 11.18 | 0.76 | 19.24 |
| ITR | 43.64 | 27.10 | 13.05 | 43.75 | 30.02 | 9.60 | 1.82 | 23.23 | 29.433 | 2.25 | 0.05 | 11.76 |
| MATE | 65.10 | 46.77 | 39.53 | 58.03 | 57.46 | 41.41 | 36.25 | 47.92 | 57.42 | 38.28 | 33.65 | 50.20 |
| AMGC | 70.05 | **54.72** | 47.81 | **67.32** | 65.62 | 45.77 | 41.85 | 59.85 | 43.83 | 29.89 | 18.46 | 46.90 |
| **Ours** | **71.16** | 50.47 | **48.79** | 63.34 | **69.42** | **49.56** | **44.72** | **62.96** | **63.22** | **43.06** | **37.16** | **59.97** |
| **CITESEER** DAEGC | 43.92 | 19.42 | 17.67 | 41.04 | 38.22 | 13.99 | 12.22 | 36.18 | 26.61 | 5.46 | 1.91 | 24.14 |
| GDCL | 61.88 | 35.10 | 34.86 | 55.31 | 40.81 | 17.27 | 13.69 | 38.23 | 21.53 | 1.13 | 0.05 | 7.26 |
| DIMVC | 54.44 | 28.37 | 27.20 | 48.69 | 35.43 | 12.63 | 8.24 | 33.62 | 20.28 | 1.04 | 0.01 | 7.04 |
| SAT | 60.83 | 32.15 | 31.84 | 53.21 | 38.84 | 16.21 | 12.06 | 37.34 | 21.89 | 1.56 | 0.13 | 8.39 |
| SVGA | 48.39 | 25.88 | 24.69 | 44.46 | 40.30 | 18.54 | 15.75 | 34.39 | 38.89 | 15.25 | 13.94 | 32.30 |
| ITR | 41.81 | 20.88 | 11.20 | 44.67 | 27.56 | 9.60 | 1.71 | 27.65 | 21.31 | 0.66 | 0.04 | 12.35 |
| MATE | 64.14 | 34.99 | 36.64 | 58.06 | 57.86 | 29.65 | 29.43 | 54.71 | 48.06 | 23.99 | 19.36 | 48.03 |
| AMGC | **68.28** | **40.37** | **42.11** | **60.66** | 62.78 | 33.85 | 35.58 | **59.16** | 32.10 | 14.14 | 4.73 | 30.37 |
| **Ours** | 66.91 | 38.98 | 39.78 | 59.25 | **63.42** | **34.05** | **36.18** | 58.90 | **50.13** | **23.83** | **21.00** | **49.50** |
| **AMAP** DAEGC | 58.48 | 48.62 | 37.34 | 53.12 | 49.04 | 37.92 | 26.02 | 46.45 | 36.04 | 20.37 | 10.35 | 30.89 |
| GDCL | 40.03 | 26.62 | 14.68 | 35.25 | 31.43 | 14.23 | 8.93 | 27.25 | 20.84 | 5.63 | 2.54 | 18.65 |
| DIMVC | 44.34 | 31.26 | 19.36 | 40.43 | 33.32 | 17.35 | 10.43 | 30.24 | 24.14 | 7.32 | 5.13 | 21.45 |
| SAT | 55.35 | 44.65 | 38.25 | 47.24 | 54.48 | 44.10 | 32.16 | 45.11 | 33.25 | 18.15 | 8.53 | 27.34 |
| SVGA | 56.6 | 50.41 | **41.07** | 45.27 | 49.05 | 41.06 | 20.10 | 36.06 | 46.35 | 33.56 | 18.48 | 29.18 |
| ITR | 41.97 | 34.41 | 11.02 | 38.37 | 35.64 | 21.60 | 2.96 | 32.07 | 27.26 | 6.20 | 0.17 | 13.46 |
| MATE | 57.88 | 48.98 | 26.31 | 49.78 | 54.36 | 48.19 | 27.19 | 46.94 | 54.37 | 48.19 | 27.20 | 46.94 |
| AMGC | | OOM | | | | OOM | | | | OOM | | |
| **Ours** | **57.89** | **53.18** | 31.76 | **53.87** | **62.90** | **55.33** | **36.28** | **56.50** | **56.30** | **52.03** | **32.94** | **50.21** |
| **AMAC** DAEGC | 37.63 | 20.34 | 15.16 | 25.35 | 28.95 | 14.14 | 8.27 | 21.69 | 19.24 | 0.75 | 0.05 | 8.35 |
| GDCL | 46.20 | 26.37 | 22.21 | 28.62 | 35.63 | 18.79 | 13.76 | 23.86 | 21.46 | 1.25 | 0.74 | 10.45 |
| DIMVC | 43.86 | 24.89 | 20.25 | 25.46 | 33.25 | 14.86 | 10.57 | 21.24 | 20.13 | 1.34 | 0.83 | 9.64 |
| SAT | 48.63 | 29.54 | 25.35 | 30.15 | 36.84 | 19.32 | 14.23 | 24.23 | 22.02 | 3.21 | 1.46 | 12.57 |
| SVGA | 45.61 | 33.41 | 23.37 | 24.04 | 45.35 | 29.93 | 22.83 | 23.86 | 35.59 | 10.58 | 8.27 | 13.51 |
| ITR | 39.16 | 30.11 | 14.10 | 28.91 | 34.94 | 18.93 | 23.23 | 22.80 | 36.54 | 5.20 | 0.30 | 10.91 |
| MATE | 43.27 | 34.05 | 20.21 | 21.61 | 48.14 | 36.11 | 21.37 | 28.92 | 41.99 | 28.07 | 18.17 | 20.07 |
| AMGC | | OOM | | | | OOM | | | | OOM | | |
| **Ours** | **60.78** | **46.75** | **42.95** | **51.48** | **58.44** | **46.18** | **38.04** | **44.09** | **55.03** | **36.14** | **29.41** | **41.02** |

Table 2: The ablation study results of R-AMGC with a missing rate of 0.6. The best results are highlighted in bold, while the second-best are underlined.

| Datasets | | | | | CORA | | | | CITESEER | | |
|---|---|---|---|---|---|---|---|---|---|---|---|
| $\mathcal{R}_e$ | $\mathcal{L}_{str}$ | $\mathcal{L}_{att}$ | $\mathcal{L}_{clu}$ | ACC | NMI | ARI | F1 | ACC | NMI | ARI | F1 |
| ✗ | ✓ | ✓ | ✓ | 57.42 | 39.32 | 30.62 | 53.92 | 52.99 | 23.52 | 22.59 | 50.04 |
| ✓ | ✗ | ✓ | ✓ | 68.09 | 48.47 | 43.00 | 61.65 | 62.00 | 33.13 | 34.31 | 57.49 |
| ✓ | ✓ | ✗ | ✓ | 67.76 | 46.64 | 42.36 | 61.30 | 51.42 | 22.55 | 21.34 | 48.71 |
| ✓ | ✓ | ✓ | ✗ | 65.17 | 47.22 | 42.74 | 58.44 | 62.45 | 32.62 | 34.73 | 58.06 |
| ✓ | ✓ | ✓ | ✓ | **69.42** | **49.56** | **44.72** | **62.96** | **63.42** | **34.06** | **36.18** | **59.90** |

CORA and CITESEER. The missing attribute ratio was progressively increased from 0.1 to 0.9, and the resulting clustering ACC and NMI were tracked, as depicted in Fig. 2.

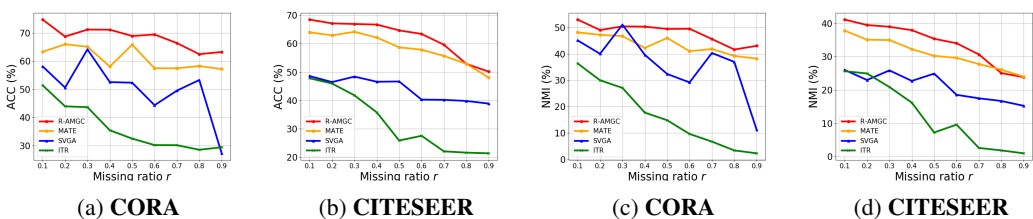

Figure 2: Performance comparison among four methods with different attribute-missing ratios

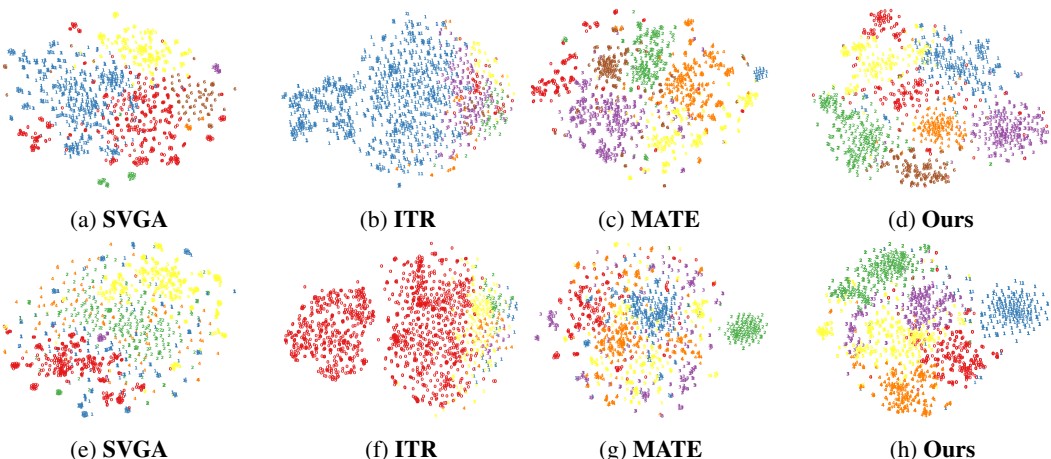

| (a) **SVGA** | (b) **ITR** | (c) **MATE** | (d) **Ours** |
|---|---|---|---|

| (e) **SVGA** | (f) **ITR** | (g) **MATE** | (h) **Ours** |
|---|---|---|---|

Figure 3: The visualization of node representation using the t-SNE algorithm, with a missing rate of 0.6, and the upper row and lower row represent the visualization for CORA and CITESEER, respectively.

### 4.5 VISUALIZATION ANALYSIS

To comprehensively demonstrate the efficacy of the proposed R-AMGC method, we conduct a visualization analysis using t-SNE Van der Maaten & Hinton (2008) on the CORA and CITESEER datasets with a missing rate of $r = 0.6$, as illustrated in Fig. 3. This approach allows for an intuitive comparison of the clustering performance across various methods, including SVGA, ITR, and MATE. Notably, our method achieves a more distinct separation of node embeddings, even in the presence of substantial missing attributes.

## 5 CONCLUSION

In this paper, we propose a Rectified Attribute-missing Graph Clustering, termed R-AMGC. Under the network framework, we employ a more flexible way of imputation, make the unknown attributes learnable, and generate other views via augmentation. Moreover, we design a novel rectification method inspired by the multi-head attention mechanism to tackle the problem of embedding distortion caused by missing nodes. Furthermore, we impose the triple constraint on the network to achieve the end-to-end learning and to get a more clustering-friendly embedding. The experimental results highlight the effectiveness and superiority of our approach, particularly in scenarios where a significant portion of node attributes are missing, showcasing its outstanding performance.

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

## A APPENDIX

You may include other additional sections here.

