# OpenReview forum: "Rectified Attribute-Missing Graph Clustering"
_ICLR.cc/2026/Conference — ICLR 2026 Conference Withdrawn Submission_

### Official Review · Reviewer_jAB7 · 2025-10-17

**Soundness:** 2
**Presentation:** 2
**Contribution:** 1
**Rating:** 2
**Confidence:** 3

**Summary:**

The paper introduces R-AMGC, an end-to-end graph clustering framework tailored to handle missing node attributes. The approach builds on two key components: (1) a rectification module using multi-head attention to reduce embedding distortion from missing nodes, and (2) a triple constraint loss that enforces consistency across graph structure, attribute reconstruction, and clustering assignments.

**Strengths:**

* Clear problem motivation: The work correctly identifies a practical challenge and positions itself as an end-to-end solution rather than a two-stage approach

* Modular design: The pipeline is structurally neat and conceptually easy to follow, which could make it extensible.

* Comprehensive evaluation setup: Experiments include multiple missing rates and datasets, with ablation studies on key components, giving some evidence of contribution isolation

* Readable methodology: Compared to some GNN clustering works, the mathematical formulation is laid out systematically, making the implementation conceptually accessible.

**Weaknesses:**

* Lack of conceptual novelty: Nearly every element of the method is a direct reuse of standard techniques (contrastive learning, heat kernel diffusion, GCN encoder, multi-head attention, reconstruction losses, and modularity-based clustering). The rectification module is essentially attention-weighted message passing with a fixed decay factor, and the triple constraint is just a weighted sum of existing loss functions. There’s no fundamentally new algorithmic insight here.

* Shallow technical depth: The rectification mechanism is described as if novel, but it's essentially GAT with a fixed $\lambda$ decay on masked nodes, which is neither theoretically analyzed nor empirically compared to simpler baselines (e.g., weighted neighbor aggregation or masking strategies).

* No theoretical or empirical justification for design choices.

* The core idea is essentially a combination of existing standard components, with minimal novelty and no rigorous justification. The empirical results, while not bad, do not show compelling or robust gains.

**Questions:**

Please see the above weaknesses

---

### Official Review · Reviewer_EB6s · 2025-10-29

**Soundness:** 3
**Presentation:** 2
**Contribution:** 2
**Rating:** 2
**Confidence:** 5

**Summary:**

This paper proposes a Rectified Attribute-Missing Graph Clustering (AMGC) method. The authors design a rectification mechanism inspired by the multi-head attention mechanism to address embedding distortion caused by missing node attributes. The rectified embeddings are then decoded to reconstruct structural and attribute information. Additionally, a clustering consistency loss is employed to ensure alignment between the graph structure and clustering results, forming an end-to-end learning framework. Extensive experiments under different missing ratios are conducted to validate the model’s performance.

**Strengths:**

1. The paper evaluates model performance under different levels of missing attributes, providing empirical evidence for the proposed approach.

2. The manuscript is clearly structured and offers a detailed overview of prior work on attribute-missing graph learning.

**Weaknesses:**

1. The paper lacks clear innovation. The rectified structure and several loss functions are largely derived from existing studies, making the proposed method appear as a straightforward extension to the missing-attribute scenario.

2. The experimental setup is relatively low-end. A higher-capacity environment (e.g., with 24 GB GPU memory) should have been adopted to ensure smooth operation of the AMGC and baseline algorithms.

3. All datasets used in the experiments are small-scale. It remains unclear whether the proposed method can be effectively applied to large-scale datasets (e.g., those with more than 10,000 samples).

4. The paper contains numerous minor errors—for instance, the first word after Equation (17) should be lowercase, and some spaces are missing.

5. In Equation (21), all loss terms appear to have the same scale. The authors should explain why no balancing coefficients were introduced to control their relative magnitudes.

**Questions:**

Please see the Weakness

---

### Official Review · Reviewer_4D4q · 2025-10-30

**Soundness:** 2
**Presentation:** 3
**Contribution:** 2
**Rating:** 2
**Confidence:** 5

**Summary:**

This paper proposes Rectified Attribute-Missing Graph Clustering (R-AMGC), an end-to-end framework for deep graph clustering under attribute-missing scenarios. The method introduces learnable imputation for missing attributes, an embedding rectification module based on multi-head attention to mitigate representation distortion, and a triple constraint loss to jointly reconstruct structure, attributes, and clustering consistency.

**Strengths:**

- The model provides a unified end-to-end solution integrating imputation, embedding learning, and clustering optimization.

- The introduction of an embedding rectification mechanism addresses the bias caused by missing node attributes.

- Experimental evaluations include multiple datasets, ablation studies, and visualization analysis.

**Weaknesses:**

- There is no clear explanation of how the model avoids overfitting or ensures the accuracy of the imputed attributes.
﻿
- The rectification matrix P and the attention-based correction process lack quantitative validation; the reliability of the “corrected” embeddings remains unclear.
﻿
- The computational complexity is not analyzed; both multi-head attention and multiple reconstruction losses could significantly increase training cost for large-scale graphs.
﻿
- The scalability of R-AMGC to large datasets is not demonstrated or discussed.
﻿
- Although the triple constraint module combines reconstruction and clustering consistency, its optimization stability and convergence properties are not theoretically analyzed.
﻿
- The comparison baselines do not include the most recent 2025 graph imputation and clustering methods, which may weaken the claim of superiority.

**Questions:**

- How does the proposed learnable imputation ensure that the filled attributes accurately approximate the true values rather than introducing noise?
﻿
- What mechanisms are employed to verify or guarantee the correctness of the rectified P matrix during training?
﻿
- How does the proposed method perform on large-scale datasets, and what is the expected computational and memory complexity with increasing graph size?
﻿
- What is the asymptotic time and space complexity of the embedding rectification module and triple constraint optimization, especially in relation to node and edge counts?

---

### Official Review · Reviewer_A3nz · 2025-10-31

**Soundness:** 2
**Presentation:** 2
**Contribution:** 2
**Rating:** 2
**Confidence:** 3

**Summary:**

This paper proposes R-AMGC, an end-to-end framework for attribute-missing graph clustering. The approach introduces two augmented attribute views, including one with learnable missing attributes, and employs a multi-head attention-based embedding rectification module to alleviate embedding distortion caused by missing attributes. A triple-constraint loss jointly optimizes structure reconstruction, attribute reconstruction, and clustering consistency. Experiments on four benchmark graph datasets demonstrate that R-AMGC achieves superior performance compared to several state-of-the-art baselines, particularly under high missing rates.

**Strengths:**

1. The proposed embedding rectification module based on multi-head attention effectively mitigates embedding distortion arising from missing attributes.
2. The framework integrates imputation and clustering in a unified manner, reducing potential error propagation associated with multi-stage methods.
3. Experiments demonstrate that the proposed approach is robust to high levels of missing attributes.

**Weaknesses:**

1. The paper claims that existing methods perform poorly under high missing rates, but this statement is not sufficiently substantiated.
2. The rationale behind the two specific attribute augmentation strategies is not clearly justified. Including theoretical insights or empirical ablations would enhance the methodological soundness.
3. In Section 3.4.1, the model performs structure reconstruction even though the graph structure is assumed complete. The motivation for this design choice should be clarified.
4. The total loss function in Eq. (21) is defined as a simple sum of four components. Discussion on the relative weighting or adaptive balancing of these terms would help assess the robustness and sensitivity of the optimization.
5. Some baseline methods used for comparison, particularly those designed for attribute-complete settings, appear outdated. Incorporating more recent graph clustering models would improve the fairness and credibility of the evaluation.
6. In Table 1, R-AMGC underperforms on certain datasets (e.g., CITESEER with 0.3 missing rate). The paper should provide an analysis or discussion of these cases to better understand the model’s limitations.

**Questions:**

1. How do existing methods specifically fail under high missing rates? Providing more analyses would strengthen the argument.
2. What is the theoretical or empirical motivation for selecting the two particular attribute augmentation strategies for generating the two views?
3. Given that the graph structure is assumed to be complete, what is the purpose of performing structure reconstruction in Section 3.4.1?
4. Have you considered adopting a weighted or adaptive loss balancing scheme for Eq. (21)? How sensitive is model performance to the choice of these weights?
5. Some attribute-complete baselines seem outdated. Have you considered comparing with more recent methods to better demonstrate the advancement of R-AMGC?
6. For the datasets where R-AMGC performs worse than baselines (e.g., CITESEER with 0.3 missing rate), what are the potential causes? Is there any identifiable pattern in dataset characteristics or missing attribute distributions that may explain this behavior?

---

### Official Review · Reviewer_DZ5T · 2025-11-03

**Soundness:** 2
**Presentation:** 3
**Contribution:** 2
**Rating:** 4
**Confidence:** 3

**Summary:**

This paper proposes R-AMGC, an end-to-end framework for attribute-missing graph clustering. It estimates missing node attributes, learns representations with contrastive learning, and rectifies embeddings to reduce the impact of missing features before clustering. Experiments on several benchmark datasets show improved performance over existing AMGC methods, especially under high missing rates.

**Strengths:**

S1. This paper addresses the attribute-missing graph clustering setting in a unified end-to-end manner.

S2. The idea of rectifying unreliable embeddings for missing-attribute nodes is clear and improves robustness.

S3. Experimental results show performance gains under higher missing-rate scenarios.

**Weaknesses:**

W1. The novelty is somewhat limited; the method mainly combines known components (imputation, contrastive learning, GAE-style clustering).

W2. Baseline comparison is incomplete and omits stronger recent GCL/AMGC methods, weakening SOTA claims.

W3. The design appears over-engineered with multiple losses and modules, without clear evidence that each part is necessary.

W4. Limited evaluation scope - no large-scale or various graph types; scalability and generality remain unclear.

**Questions:**

Q1. What is the main technical novelty beyond combining existing imputation, contrastive, and GAE-based clustering techniques?

Q2. Can you include evaluations against stronger recent GCL/AMGC baselines to more reliably support the SOTA claim? Is it possible or not?

Q3. Can you provide ablations isolating each module and loss to confirm which components are actually necessary?

Q4. How does the proposed method perform on larger graphs and different graph types (e.g., heterophilic) to validate scalability and generality?

---

### Note · Authors · 2025-11-12

I have read and agree with the venue's withdrawal policy on behalf of myself and my co-authors.